# Verification of the Impact of Blood Glucose Level on Liver Carcinogenesis and the Efficacy of a Dietary Intervention in a Spontaneous Metabolic Syndrome Model

**DOI:** 10.3390/ijms222312844

**Published:** 2021-11-27

**Authors:** Mayuko Ichimura-Shimizu, Takeshi Kageyama, Takeshi Oya, Hirohisa Ogawa, Minoru Matsumoto, Satoshi Sumida, Takumi Kakimoto, Yuko Miyakami, Ryosuke Nagatomo, Koichi Inoue, Chunmei Cheng, Koichi Tsuneyama

**Affiliations:** 1Department of Pathology and Laboratory Medicine, Institute of Biomedical Sciences, Tokushima University Graduate School, Tokushima 770-8503, Japan; ichimura.mayuko@tokushima-u.ac.jp (M.I.-S.); c201501111@tokushima-u.ac.jp (T.K.); ogawa.hirohisa@tokushima-u.ac.jp (H.O.); sumida.satoshi@tokushima-u.ac.jp (S.S.); takumi1124_aaliyah@yahoo.co.jp (T.K.); miyakami.yuko@tokushima-u.ac.jp (Y.M.); 2Department of Molecular Pathology, Institute of Biomedical Sciences, Tokushima University Graduate School, Tokushima 770-8503, Japan; oya.takeshi@tokushima-u.ac.jp (T.O.); m.matsumoto@tokushima-u.ac.jp (M.M.); 3Laboratory of Clinical and Analytical Chemistry, College of Pharmaceutical Sciences, Ritsumeikan University, Kusatsu, Shiga 525-8577, Japan; ph0070fs@ed.ritsumei.ac.jp (R.N.); kinoue@fc.ritsumei.ac.jp (K.I.); 4Pharmacology and Histopathology, Novo Nordisk Research Centre China, Beijing 102206, China; cceg@novonordisk.com

**Keywords:** metabolic syndrome, Tsumura Suzuki Obese Diabetes mouse, blood glucose, islet of Langerhans, oligosaccharide

## Abstract

Metabolic syndrome (MS) is a risk factor for type 2 diabetes mellitus, vascular inflammation, atherosclerosis, and renal, liver, and heart diseases. Non-alcoholic steatohepatitis (NASH) is a progressive representative liver disease and may lead to the irreversible calamities of cirrhosis and hepatocellular carcinoma. Metabolic disorders such as hyperglycemia have been broadly reported to be related to hepatocarcinogenesis in NASH; however, direct evidence of a link between hyperglycemia and carcinogenesis is still lacking. Tsumura Suzuki Obese Diabetic (TSOD) mice spontaneously develop metabolic syndrome, including obesity, insulin resistance, and NASH-like liver phenotype, and eventually develop hepatocellular carcinomas. TSOD mice provide a spontaneous human MS-like model, even with significant individual variations. In this study, we monitored mice in terms of their changes in blood glucose levels, body weights, and pancreatic and liver lesions over time. As a result, liver carcinogenesis was delayed in non-hyperglycemic TSOD mice compared to hyperglycemic mice. Moreover, at the termination point of 40 weeks, liver tumors appeared in 18 of 24 (75%) hyperglycemic TSOD mice; in contrast, they only appeared in 5 of 24 (20.8%) non-hyperglycemic mice. Next, we investigated three kinds of oligosaccharide that could lower blood glucose levels in hyperglycemic TSOD mice. We monitored the levels of blood and urinary glucose and assessed pancreatic lesions among the experimental groups. As expected, significantly lower levels of blood and urinary glucose and smaller deletions of Langerhans cells were found in TSOD mice fed with milk-derived oligosaccharides (galactooligosaccharides and lactosucrose). At the age of 24 weeks, mild steatohepatitis was found in the liver but there was no evidence of liver carcinogenesis. Steatosis in the liver was alleviated in the milk-derived oligosaccharide-administered group. Taken together, suppressing the increase in blood glucose level from a young age prevented susceptible individuals from diabetes and the onset of NAFLD/NASH, as well as carcinogenesis. Milk-derived oligosaccharides showed a lowering effect on blood glucose levels, which may be expected to prevent liver carcinogenesis.

## 1. Introduction

Metabolic syndrome (MS) is an intractable disease, and it may initiate from obesity and then develop into systemic disorders, such as type 2 diabetes, hyperlipidemia, arteriosclerosis, and non-alcoholic steatohepatitis (NASH) [1,2]. Diabetes characterized with high blood glucose level has been regarded as a risk factor for NASH-related cirrhosis and hepatocellular carcinoma (HCC) [3,4,5]. Given the complexity of diabetes patients’ backgrounds and their genetical susceptibility, a question arose regarding whether a high blood glucose level could be a causal factor for liver carcinogenesis in the diabetic population. Tsumura Suzuki obese diabetic (TSOD) mice without any diet intervention spontaneously display typical phenotypes of metabolic syndrome, including severe obesity, type 2 diabetes, hyperinsulinemia, steatohepatitis, and HCC [6,7,8]. TSOD mice are a multifactorial inherited diabetes model that are inbred from closed-colony ddY mice for several generations by multiplying individuals with obesity and high urinary sugar levels [9]. The onset of pathological conditions in TSOD mice might result from multiple factors similar to those found in human diabetes. In this study, we monitored TSOD mice regarding changes in blood glucose levels, body weight, and pancreatic and liver lesions over time. Furthermore, we investigated the effect of dietary intervention with various kinds of oligosaccharide in TSOD mice in terms of lowering blood glucose levels from a young age. It has been reported that the regulation of intestinal microbiota with healthy and balanced foods could prevent the occurrence and progression of MS-related diseases [10]. Recently, increasing reports have shown evidence that oligosaccharides improve the intestinal environment and various symptoms of metabolic syndrome [11]. Our previous study also showed the treatment effect of fructooligosaccharides (FOS) for NASH in a diet-induced NASH model and an MSG-induced NASH model [12,13]. In the FOS-administered group, the level of total short-chain fatty acids (SCFAs) increased compared to the vehicle group, which indicated that FOS administration might improve intestinal microbiota. As SCFAs have been reported to prevent colonic mucosa and pancreatic beta-cells from being damaged, SCFAs have been considered as key factors for ameliorating MS and NASH [13]. Various types of oligosaccharides are on the market; however, they lack head-to-head comparison studies regarding their efficacy in NASH. In this study, we showed the protective effect of milk-related oligosaccharides (galactooligosaccharide: GOS and lactosucrose: LS) as well as FOS on lowering blood glucose level, preventing pancreatic beta-cell damage in TSOD mice.

## 2. Materials and Methods

### 2.1. Ethics Statement

All the institutional and national guidelines for the care and use of laboratory animals were followed. This study was performed in alignment with the animal experiment guidelines specified by the Institute for Animal Reproduction (Ibaraki, Japan), (Permission number: IarAW No. 2018-N565), where the rules of guidance on animal research ethics from the International Association of Veterinary Editors’ Consensus Author Guidelines on Animal Ethics and Welfare were strictly abided by.

### 2.2. Animals and Experimental Design

Experiment 1: To investigate the impact of blood glucose level on the frequency of hepatic carcinogenesis at a young age in TSOD mice.

TSOD mice fed with a murine-certified diet (MF, Oriental yeast, Tokyo, Japan) developed obesity around eight weeks of age and then spontaneously developed type 2 diabetes, hyperlipidemia, and human NASH-like steatohepatitis in sequence at around six months of age. In our previous studies, liver tumors appeared in TSOD mice after eight months of age and the frequency increased over the time; for instance, over 80% of TSOD mice showed liver tumors after 12 months of age [7]. In this study, based on blood glucose levels, male TSOD mice were divided into a hyperglycemic group (*n* = 25) and a non-hyperglycemic group (*n* = 24) at 12 weeks of age, and eight mice from each group were sacrificed at 32, 36, and 40 weeks of age for time-course analysis. Bodyweight was monitored for all experimental mice at 4, 5, 6, 12, 16, 24, 32, and 40 weeks of age. At each termination point, animals were first fasted for 12 h and then liver and pancreas were collected under anesthesia with carbon dioxide. After a gross necropsy was performed, the organs were fixed in 10% neutral buffered formalin, embedded in paraffin, and stained with hematoxylin and eosin (HE) for microscopy.

Experiment 2: To investigate the effect of dietary oligosaccharides on suppressing blood glucose level at a young age in TSOD mice.

Male TSOD mice showing hyperglycemia at 8 weeks of age and fed with a murine-certified diet (MF) were selected and randomly divided into 4 groups, with 4–5 mice per group; these mice were treated with FOS, GOS, or LS. Five mice without any treatment were employed as controls. Assessment started at 12 weeks of age, before the onset of MS symptoms such as obesity based on bodyweight, and it continued up to 24 weeks of age, when all mice sequentially developed the phenotypes of diabetes. Each group was continuously administered an MF with 5% oligosaccharide in drinking water ad libitum. Based on the water intake of TSOD mice per day, this amounted to approximately 4–6 mL; 0.005 μg of oligosaccharide per mouse per day was also administered. The general life signs of the animals were observed once a day during the study. All animals were weighed once a week throughout the study. The animals were fasted for 12 h before being sacrificed, and liver and pancreas were collected under anesthesia with carbon dioxide. After fixation in 10% neutral buffered formalin, representative liver slices and a maximum cut surface of pancreas were embedded in paraffin and then cut at 5 μm thickness for morphological assessment by HE staining.

### 2.3. Blood Glucose Level Measurement

Blood glucose levels were measured between 10 am and 12 pm at 6, 12, 16, 24, 32, and 40 weeks of age in experiment 1 and once a week in experiment 2 using a Stat Strip Express 900 device (Nova Biomedical, Waltham, MA, USA).

### 2.4. Urine Sugar Measurement

In experiment 2, the urine sugar level of animals was measured once a week, starting from 11 weeks and up to 24 weeks of age. Fresh urine of the animals was collected and then applied on Uro-paper IIIG (Eiken Chemical, Tokyo, Japan). The scores of urine sugar levels from 0 (−) to 5 (++++) were determined by following the instructions of the manufacturer.

### 2.5. Glucose Tolerance Test

In experiment 2, glucose tolerance tests were performed on the control, GOS, and LS groups at 23 weeks of age. After fasting for 17 h, fasting blood glucose levels were measured using a Stat Strip Express 900 device (Nova Biomedical, Waltham, MA, USA). Subsequently, 150 mg/mL of glucose was applied orally and blood glucose levels were measured at 60 and 120 min post-administration in examined animals.

### 2.6. Morphological Assessment of the Liver

In experiment 1, after fixing the liver with 10% neutral buffered formalin, whole livers were cut through at 2 mm intervals and examined grossly for tumor appearance. All sections with tumors were embedded in paraffin and thin-cut to 2 μm for microscopic examination. Routine HE staining and immunostaining using rabbit-polyclonal anti-glutamine synthetase (GS) antibody (rab73593, Abcam plc, Cambridge, UK) were performed to identify liver tumors. In experiment 2, for NASH evaluation, hepatic steatosis, lobular inflammation, ballooning, and fibrosis were scored according to NAS scoring [14]. Since no liver tumor was found at 24 weeks of age in the previous report, we only performed histological assessments for liver tumors after 24 weeks.

### 2.7. Morphological Assessment of the Pancreas

In both experiment 1 and experiment 2, the destructive degree of islets of Langerhans was evaluated at three levels: no change, mild change, and severe change, based on the beta-cell expression by immunostaining of insulin (rabbit monoclonal, EP125, Epitomics, Burlingame, CA, USA). In addition, the area of islets of Langerhans was measured by image analysis software (CellSens Standard, Olympus, Tokyo, Japan).

### 2.8. Analysis of mRNA Expression in Liver

In experiment 2, we analyzed the expression of hepatic lipid metabolism-related genes, such as carnitine palmitoyltransferase (*Cpt-1a*), fatty acid synthase (*Fasn*), fatty acid transport protein 5 (*Fatp5*), cluster of differentiation 36 (*CD36*), carbohydrate response element binding protein (*Chrebp*) and microsomal triglyceride transfer protein (*Mttp*) and glycogenesis-related genes including glucose-6-phosphatase (*G6pase*), phosphoenolpyruvate carboxykinase 1 (*Pepck1*) and *Pepck2*. Total RNA was isolated using a bead crusher (Micro Smash MS-100, TOMY, Tokyo, Japan) and a RNeasy Mini Kit (Qiagen, Hilden, Germany). cDNA was synthesized using ReverTra Ace qPCR RT Master Mix (Toyobo, Osaka, Japan) and then stored at −20 °C until we performed a StepOne real-time PCR (Applied Biosystems, Waltham, MA, USA). The sequences of the synthetic primers are listed in Table 1. The relative ratio of each gene expression level to that of beta-actin was calculated using the 2^−^^ΔΔCt^ method and expressed as fold of levels in controls.

### 2.9. Analysis of Plasma SCFAs

Plasma levels of acetate, propionate, butyrate, isobutyrate, 2-methylbuthyrate, and capronate of mice were examined using UPLC-ESI-MS/MS. Reference materials were purchased from Wako Pure Chemical Co. (Tokyo, Japan). Internal standards (IS), PA- deuterium (d) 6, BA-d5, VA-d9, and CA-d11 were obtained from Sigma-Aldrich Co. (St. Louis, MO, USA) and CDN Isotopes Co. (Pointe-Claire, QC, Canada).

### 2.10. Statistical Analyses

Statistical difference compared among groups was analyzed by Mann–Whitney’s U test and ANOVA with Dunnett’s post hoc analysis.

## 3. Results

1: Experiment 1: blood glucose level impacted the frequency of hepatic carcinogenesis at a young age in TSOD mice

### 3.1. Pathological Characteristics of Islets of Langerhans

In the pancreas of TSOD mice that showed persistent hyperglycemia, the invasion of exocrine cells into islets of Langerhans was observed at 32 weeks of age. This was considered a consequence of pathologically swollen islets of Langerhans due to beta-cells’ struggling with compensation and decompensation [15] (Figure 1A–C). Reserved endocrine cells were positively stained by insulin (Figure 1D–F). Regarding immunostaining with insulin, Figure 1D shows swelled islets of Langerhans without the invasion of exocrine glands at 32 weeks of age; Figure 1E shows a mild reduction in beta-cell amounts in the islets of Langerhans at 32 weeks of age; and Figure 1D shows severe deletion of beta-cells in the islets of Langerhans at 36 weeks of age.

### 3.2. Changes in Blood Glucose Level and Body Weight between Hyperglycemic and Non-Hyperglycemic TSOD Mice

Figure 2 shows the changes in blood glucose levels and bodyweight between the hyperglycemic and non-hyperglycemic mice. In the group of hyperglycemic TSOD mice showing hyperglycemia at 12 weeks, blood glucose levels increased up to 16 weeks and then gradually decreased; however, significantly higher blood glucose levels were maintained up to 40 weeks of age compared to those in the non-hyperglycemic group (Figure 2A). Likewise, bodyweight was significantly heavier in the hyperglycemic group from 8 to 32 weeks of age compared to the non-hyperglycemic group; however, at the 40-week point, all mice maintained the same levels (Figure 2B).

### 3.3. Frequency of Liver Tumors

Liver tumors were confirmed in four out of eight (50%) animals at 32 weeks of age in the hyperglycemic group, but these were not found in the non-hyperglycemic group (Table 2 and Figure 3). At the termination point of 40 weeks, liver tumors appeared in 18 of 24 (75%) hyperglycemic mice, but only in 5 of 24 (20.8%) non-hyperglycemic mice. Moreover, a significantly greater number of liver tumors was found in hyperglycemic mice compared to non-hyperglycemic mice (*p* < 0.01, Figure 3A HE staining at low magnification, Figure 3B HE stain at high magnification, and Figure 3C GS immunostaining).

### 3.4. Changes in Islets of Langerhans in the Pancreas

The areas of islets of Langerhans in the hyperglycemic group were significantly increased compared to the non-hyperglycemic group at 32, 36, and 40 weeks of age, respectively (Figure 4A). This phenomenon might be interpreted as a condition of insulin resistance in beta-cells compensating their function by increasing their size and sequentially distorting the structure of Langerhans. The mice with severe morphological distortions of islets of Langerhans showed higher blood glucose levels compared to those with non or mildly morphological change in islets of Langerhans (Figure 4B). Therefore, the severe morphological changes in islets of Langerhans might reflect beta-cell stress due to persistent hyperglycemia (Table 3). The incidence of liver tumors also showed a positive correlation with morphological abnormalities in islets of Langerhans (Figure 4C).

2: Experiment 2: Dietary oligosaccharides suppressed blood glucose level at a young age in TSOD mice

### 3.5. Changes in Bodyweight and Glucose Levels

All animals gained bodyweight during the study period, with no significant differences observed between groups (Figure 5A). The bodyweight of all animals at 24 weeks tended to be lower than that at 23 weeks, which was considered a result of fasting for the glucose tolerance test. The blood glucose levels of the GOS group were lower than those of the controls and the group fed with LS from 14 weeks of age up to the endpoint of the study (Figure 5B). Blood glucose levels were significantly decreased in mice fed with GOS than in those fed with LS at 19, 23, and 24 weeks (*p* < 0.05, Figure 5B). Among the mice fed with LS, three animals showed low blood glucose levels from 13 to 20 weeks, and one animal showed a rebound in blood glucose levels at 21 and 22 weeks. The blood glucose levels of mice fed with FOS were higher than in controls up to 15 weeks of age; they then decreased in a similar manner to controls (Figure 5B). Urine sugar was undetectable in three of the GOS-fed animals. Overall, the urine sugar levels of GOS-fed mice were significantly lower than those of controls from 16 to 23 weeks of age (*p* < 0.05, Figure 5C). Urine sugar was undetectable in one of the LS-fed animals at 11 weeks and then again at 16–24 weeks of age. Therefore, the overall urine sugar levels of LS-fed mice were significantly lower than those of controls at 16, 19, and 23 weeks (*p* < 0.05, Figure 5C). In parallel to blood glucose levels, the urine sugar levels of FOS-fed mice were higher up to 20 weeks of age compared to those of mice fed with GOS and LS; these then decreased afterwards (Figure 5C). Blood glucose levels of mice fed with both GOS and LS were lower than those of controls at 60 and 120 min post-administration of the oral glucose tolerance test, and a statistical difference was found in LS-fed mice compared to the controls (*p* < 0.05) (Figure 5D).

### 3.6. Histopathological Findings in the Liver

In control animals, lobular inflammation was occasionally observed, and hepatic apoptotic (acidophilic) bodies were also found in random locations. Although mild steatosis was seen in four out of five cases (80%), there was no ballooning degeneration observed in any of the cases. Other than one out of four animals fed with FOS showing mild steatosis, no other oligosaccharides-administrated animals showed considerable steatosis (Table 4 and Figure 6).

### 3.7. mRNA Expression of Lipid Metabolism-Related Genes in the Liver

It is well known that Fasn is involved in de novo synthesis of fatty acid, Fatp5 and CD36 is related to fatty acid uptake, Cpt-1a participates in fatty acid beta-oxidation, and Mttp plays a role in triglyceride excretion, Chrebp is a transcription factor in the carbohydorate and lipid synthesis, and Pepck and G6pase are the enzymes responsible for glycogenesis in the liver. In this study, we unexpectedly found no significant difference in the mRNA level of the above genes between the control group and the oligosaccharide-administered groups (Figure 7). Either GOS or LS administration showed mild suppression of fatty liver, however, the underlying mechanism could not be explained regarding how these genes were involved in lipid and carbohydrate metabolism.

### 3.8. Histopathological Findings in the Pancreas

Unremarkable morphology change was observed in the exocrine gland of the controls; however, various degrees of extension of the exocrine gland into the islets of Langerhans were observed (Figure 8). In comparison, significant improvement in the integrality of islets of Langerhans were observed in the mice fed with GOS and LS, but not in the mice fed FOS (Table 5, Figure 8).

### 3.9. Quantitation of Plasma SCFAs

There were no significant differences in the plasma level of examined short chain fatty acids among the four groups (Table 6). In the FOS and GOS groups, all the short-chain fatty acids retrieved showed the lower trend compared to the control group, in contrast, propionate and butyrate in the LS group showed a higher trend than that in the control group, indicating differences in the changes in the bacterial flora depending on the oligosaccharide type might be the explanation.

## 4. Discussion

TSOD mice spontaneously develop obesity, type 2 diabetes, hyperlipidemia, NASH, and eventually develop a high frequency of liver tumors after 18 months of age [7,16]. A TSOD mouse with a genetic breeding background but no special diet intervention exhibits a characterized model mimic of human metabolic syndrome. In addition, although TSOD mice are genetically homologous, significant individual variations appear with various pathological phenotypes, with some showing hyperglycemia and some not even doing so at eight weeks of age. In this study, TSOD mice with and without hyperglycemia at 12 weeks were parallelly assessed for metabolic syndrome over the study period, and the incidence of liver tumors was also examined. It was found that TSOD mice without hyperglycemia at 12 weeks of age constantly maintained low blood glucose levels up to the end of the study, and their bodyweight was significantly lower than that of mice with hyperglycemia. In our study, we found that liver tumors occurred in about half of the hyperglycemic TSOD mice at 32 weeks of age; in contrast, no tumors were found in the non-hyperglycemic group at this age. That said, liver tumors were confirmed in the non-hyperglycemic mice with at a rate of 12.5% at 36 weeks of age and up to 50% at 40 weeks of age. These data suggested that the onset of hepatic carcinogenesis was delayed in the non-hyperglycemic TSOD mice; in other words, hyperglycemia might accelerate hepatic carcinogenetic signaling in TSOD mice due to their genetic susceptibility. These findings are well aligned with current studies showing a strong association between diabetes and hyperglycemia with carcinogenesis in various organs [17,18,19].

Recently, accumulating studies had shown evidence that the disturbance of intestinal microbiota might play an important role in metabolic diseases [20]. Various attempts have been undertaken to change intestinal flora, such as prebiotics, probiotics, and combined synbiotics [21]. It has been reported by many studies that oligosaccharides lower blood glucose levels with different modes of action speculated among the types of oligosaccharides such as immune response [22]. In order to focus on functional food ingredients that could be safely taken by patients in their diet, we chose to add oligosaccharides into a murine-certified diet based on their therapeutic effects on metabolic syndrome and capacity to improve intestinal bacterial flora [12,23,24]. As expected, milk-derived oligosaccharides alleviated blood and urine glucose levels and restored the integrality of pancreatic islets of Langerhans in TSOD mice. These results implied that a direct or indirect beta-cell protection was exerted by the administration of milk-derived oligosaccharides. Since mild NASH-like liver pathology was found in the control group, the therapeutic window was not large enough to evaluate the efficacy of oligosaccharides against liver inflammation and fibrosis, at least not in this study. TSOD mice comprise a spontaneous model, and strong liver fibrosis could not be confirmed. We considered that the present results are insufficient for evaluating the anti-fibrotic potential of oligosaccharides. We recently developed a diet-induced NASH model showing strong liver fibrosis [24]. An additional study using a severe liver fibrosis model is required.

As for metabolic disturbance, improvements in fatty degeneration were observed in the oligosaccharide-treated groups. To unveil the underlying metabolic actions in the liver, we examined lipid-metabolism related genes such as Cpt-1a, Fasn, Fatp5, CD36, ChREBP, Mttp, and glycogenesis-related genes including G6pase, Pepck1, Pepck2. However, all genes examined in the present study did not reveal the evidence of oligosaccharide-induced fatty liver suppression. One possibility for lacking significance between the treatment groups vs. controls in this study was considered that the degree of hepatic fatty degeneration was too mild in the control group. As mentioned above, the advantage of TSOD mice is their spontaneous development of various metabolic syndrome, however, the disadvantage is the NASH-like pathogenic phenotypes less severe than other NASH models. Therefore, these metabolic genes in the diet-induced NASH model will be further investigated.

Regarding the morphological changes in islets of Langerhans in the pancreas and their increased areas in hyperglycemic TSOD mice, it is likely that these changes reflect the compensatory swelling of beta-cells under insulin resistance; this is because of the exocrine gland’s invasion into islets and beta-cell fragmentation clustering coincidently. Moreover, the degree of islet destruction was strongly correlated with the occurrence of liver tumors. Therefore, our results propose a novel clinic perspective insofar as the risk of hepatic carcinogenesis could be predicted by the distortion degree of islets of Langerhans at a younger age. Based on the protective effects of milk-related oligosaccharides on beta-cells, we hypothesized that the potential underlying mode of action might be SCFAs produced from oligosaccharides by intestinal bacteria, but no significant changes in serum SCFA levels was observed. More sophisticated analysis on either the amount or organ-specific localization of SCFAs will be further investigated, since beta-cells have SCFAs receptors [25,26], and several studies suggest the effect of propionic acid and butyric acid on improving insulin sensitivity and preventing inflammation. [27,28,29,30].

A limitation of the present study is that we were unable to analyze factors related to metabolism and tissue damage, such as hormone and cytokine activities, over time. In the present study, only males were used in the experiments due to the severity of the condition; however, a new experimental design that takes into account the differences between males and females is necessary to consider the relationship between these factors regarding their response. Since female TSOD mice also develop metabolic syndrome and liver tumors, albeit mildly, we plan to conduct additional research using male and female TSOD mice in our next study.

Contrary to expectations from our previous study [13], we were intrigued that various items were exacerbated by FOS treatment. The mice of FOS group gained weight but did not improve blood glucose levels. Oligosaccharides are not readily absorbed in the small intestine, and it is commonly believed that they cause little weight gain even when loaded with sugar [31]. However, Liu et al. warned that short-term prebiotic administration might have adverse effects on glucose metabolism according to the results they showed that when healthy young adults were administrated FOS or GOS for 14 days, bifidobacteria increased but butyrate-producing microorganisms decreased [32]. Taken together, changes in body weight and blood glucose levels may vary among the individuals even with the same oligosaccharide administration. Therefore, the patient-center-minded treatment with oligosaccharide for clinic purpose is pivotally necessary. In the future, when planning individualized prebiotic intervention based on gut microbiota, especially for young adults, it will be necessary to select the optimal prebiotic and determine the method of administration.

In conclusion, the dietary intake of milk-related oligosaccharides could lower blood glucose levels in TSOD mice and potentially prevent the development of diabetes and the onset of hepatocarcinogenesis in genetically susceptible individuals. Unlike strenuous exercise and dietary restrictions, milk-related oligosaccharides are pure dietary elements that can be safely administered for susceptible MS populations with good compliance.

## Figures and Tables

**Figure 1 ijms-22-12844-f001:**
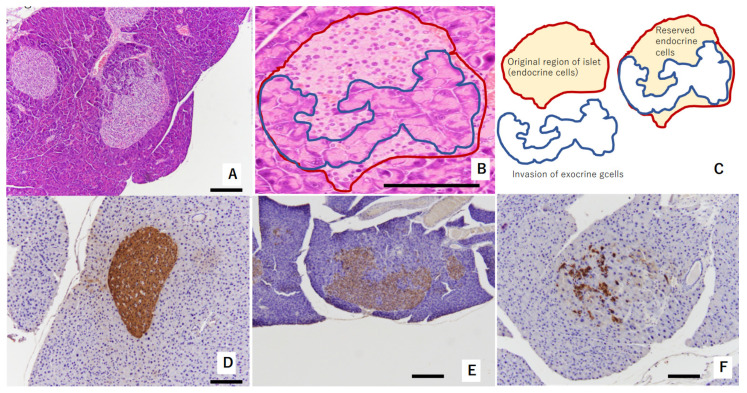
Representative figure showing morphological abnormalities of islets of Langerhans in experiment 1: (**A**,**B**) exocrine gland extends into swelled islet of Langerhans. Red circle shows original region of islet endocrine cells. The area circled by blue shows invasion of exocrine cells. These areas are shown in (**C**) as a schema. These islets were evaluated as “mild morphological change” (HE staining). (**D**) Swollen islet of Langerhans was constructed by beta-cells with abundant insulin (IHC of insulin). This islet was evaluated as “no change”. (**E**) Exocrine gland extended into swollen islet of Langerhans. Swollen islet of Langerhans was constructed by mixture of beta-cells with weak insulin (IHC of insulin) and exocrine gland. This islet was evaluated as “mild morphological change”. (**F**) Exocrine gland severely extended into swollen islet of Langerhans. Scattered beta-cells (IHC of insulin) were observed within originally swollen islet of Langerhans. This islet was evaluated as “severe morphological change”. Scale bar: 200 μm.

**Figure 2 ijms-22-12844-f002:**
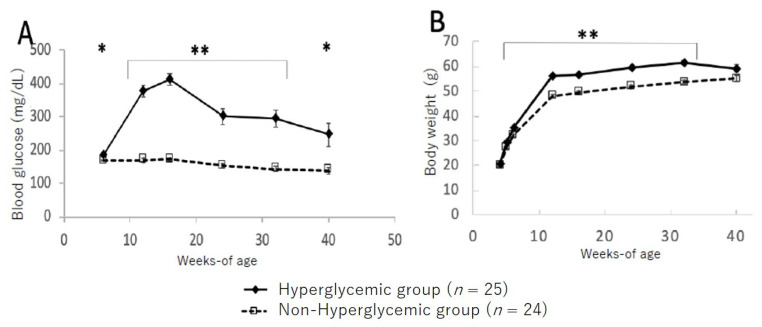
Changes in blood glucose level (**A**) and bodyweight (**B**) in hyperglycemic group and non-hyperglycemic group. Values are expressed as means ± SD. * *p* < 0.05, ** *p* < 0.01 (Dunnett’s post hoc analysis).

**Figure 3 ijms-22-12844-f003:**
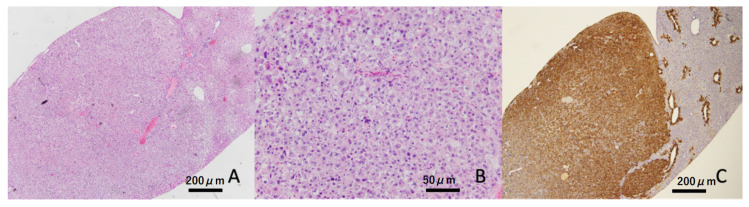
Representative image of the liver tumor. Liver tumor was constructed by atypical hepatocytes (HE staining). These tumors showed diffuse expressions of glutamine synthetase (GS). Scale bar: (**A**,**C**) 200 μm, (**B**) 50 μm.

**Figure 4 ijms-22-12844-f004:**
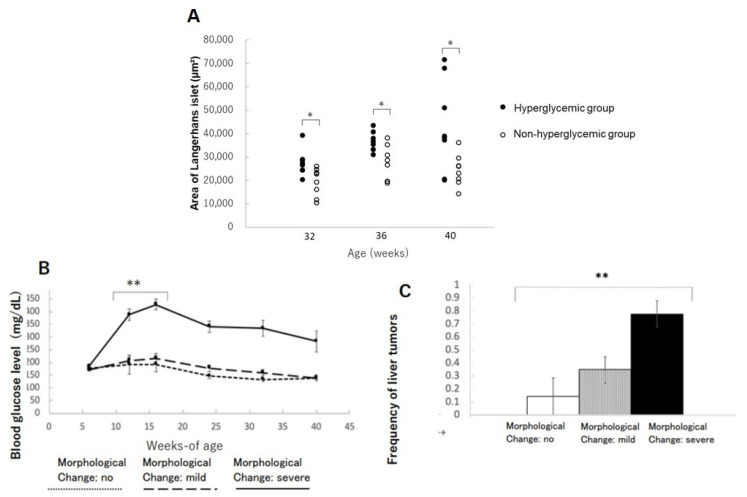
Changes in the area of islets of Langerhans in the pancreas: (**A**) changes in the area of islets of Langerhans in hyperglycemic group and non-hyperglycemic group over time. * *p* < 0.05. (**B**) Severity of morphological changes of islet of Langerhans and blood glucose level over time. ** *p* < 0.01. (**C**) Correlation between morphological abnormalities of islet of Langerhans and liver tumor incidence. ** *p* < 0.01.

**Figure 5 ijms-22-12844-f005:**
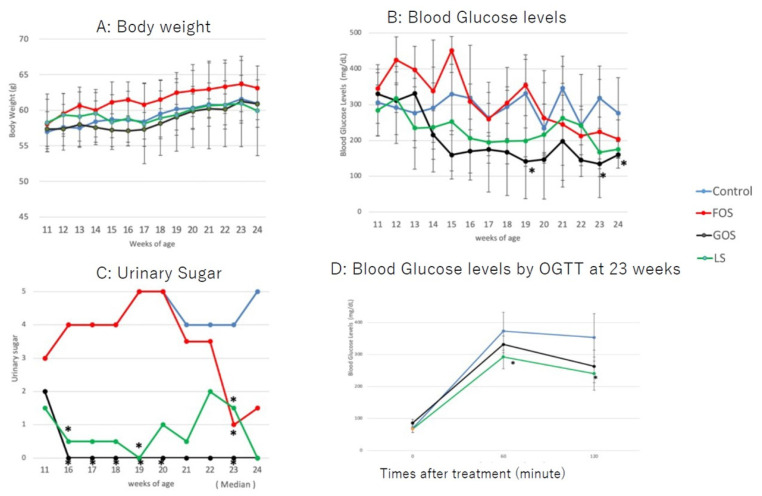
Changes in body weight (**A**), blood glucose level (**B**), and urine glucose (**C**) in the experimental groups. (**D**) Blood glucose levels by OGTT at 23 weeks. Values are expressed as means ± SD. * *p* < 0.05 vs. control (Dunnett’s post hoc analysis).

**Figure 6 ijms-22-12844-f006:**
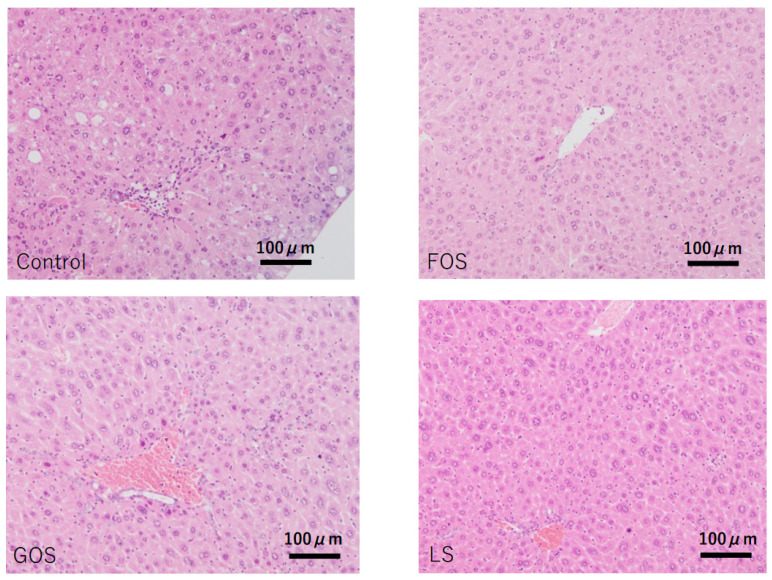
Characteristic pathological features of the liver in each group. Control: mild fatty changes as well as mild necro-inflammatory changes were observed. FOS, GOS, LS: no significant changes, including steatosis, were observed. (HE staining, scale bar: 100 μm).

**Figure 7 ijms-22-12844-f007:**
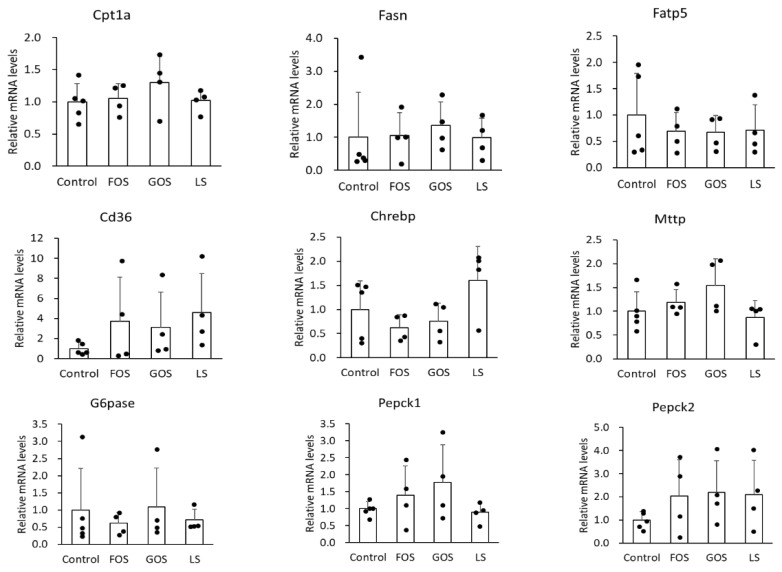
Relative mRNA levels of relate genes involved in lipid metabolism (Cpt1a, Fasn, Fatp5, CD36, Chrebp, and Mttp) and glycogenesis-related (G6pase, Pepck1, and Pepck2) genes in the liver. Values are expressed as means ± SD.

**Figure 8 ijms-22-12844-f008:**
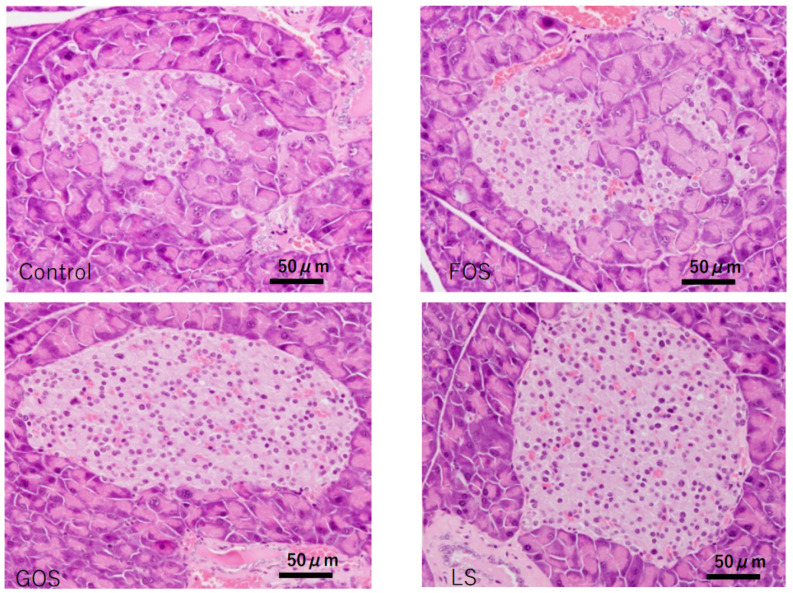
Characteristic pathological features of the islets of Langerhans in the pancreas of each group in experiment 2. In the control and FOS groups, most of the islets showed swelling and degeneration, while there were no destructive changes observed in most of the islets in GPS and LS groups (HE staining, scale bar: 50 m).

**Table 1 ijms-22-12844-t001:** Sequences of the primers synthesized for this study.

Genes	Forward (5′→3′)	Reverse (5′→3′)
*β-actin*	CATCCGTAAAGACCTCTATGCCAAC	ATGGAGCCACCGATCCACA
*Cd36*	TCCCAGAATTCTCAGCTGCTCC	CACATTTCAGAAGGCAGCAACTTC
*Chrebp*	CGACACTCACCCACCTCTTC	TTGTTCAGCCGGATCTTGTC
*Cpt-1a*	TCCTGAAGGAGGTACTGTCTG	CATAGCCGTCATCAGCAACC
*Fasn*	GACTCGGCTACTGACACGAC	CGAGTTGAGCTGGGTTAGGG
*Fatp5*	GGGTGTGAGGGTAAGGTTGG	CCAGGGAATCCTGGATACGG
*G6pase*	CCAACGTATGGATTCCGGTGT	GCAAGGTAGATCCGGGACAG
*Mttp*	TGCTGTCCATTGGGGAACTC	CGACGGATCATTTTGCTTGC
*Pepck1*	TGGAAGGTCGAATGTGTGGG	TAGCCCTTAAGTTGCCTTGGG
*Pepck2*	GTGGTAACTCCTTGCTGGGC	TTGGTGATGCCCAAAATCAGCAT

**Table 2 ijms-22-12844-t002:** Frequency of liver tumors in TSOD mice in experiment 1.

	32-Weeks	36-Weeks	40-Weeks
Hyperglycemic group	4/8 (50%)	8/8 (100%)	6/8 (75%)
Non-hyperglycemic group	0/8 (0%)	1/8 (12.5%)	4/8 (50%)

**Table 3 ijms-22-12844-t003:** Frequency of the damage of islets of Langerhans in TSOD mice in experiment 1.

	No	Mild	Severe
Hyperglycemic group	1/24 (4%)	5/24 (20.8%)	18/24 (75%)
Non-hyperglycemic group	6/24 (25%)	18/24 (75%)	0/24 (0%)

**Table 4 ijms-22-12844-t004:** Summary of histopathological characteristics of the liver in experiment 2.

Organ	Liver	Liver	Liver	Liver
Assessment Item	Steatosis	Lobular Inflammation	Ballooning	Fibrosis
Control-M1	1	2	0	1
Control-M2	1	2	0	1
Control-M3	0	2	0	0
Control-M4	1	2	0	0
Control-M5	1	2	0	0
FOS-M6	0	2	0	0
FOS-M7	0	2	0	0
FOS-M8	0	2	0	1
FOS-M9	1	2	0	0
GOS-M10	0	2	0	1
GOS-M11	0	1	0	0
GOS-M12	0	0	0	0
GOS-M13	0	2	0	1
LS-M14	0	0	0	0
LS-M15	0	2	0	1
LS-M16	0	1	0	1
LS-M17	0	2	0	1

Hepatic steatosis, lobular inflammation, ballooning, and fibrosis were scored according to the NAS score.

**Table 5 ijms-22-12844-t005:** Histopathologic evaluation of the damage of islets of Langerhans in experiment 2.

Group	*n*	No	Mild	Severe
Control	5	0	4	1
FOS	4	1	0	3
GOS	4	4	0	0
LS	4	4	0	0

**Table 6 ijms-22-12844-t006:** A titer of primary short-chain fatty acids in each group.

	Acetate	Propionate	Butyrate	Isobutyrate	2-Methyl Butyrate	Capronate
	(μM)	(μM)	(μM)	(μM)	(μM)	(μM)
Control-M1	475.5	2.7	0.9	3.4	2.3	1.7
Control-M2	283.9	3.5	1.4	1.9	N.D.	1.2
Control-M3	945.3	5.5	2.4	3.7	N.D.	3.0
Control-M4	589.9	5.6	2.6	2.7	1.7	2.4
Control-M5	889.8	N.D.	0.9	N.D.	N.D.	3.5
Ave (SD)	636.9 (279.3)	4.3 (1.4)	1.6 (0.8)	2.9 (0.8)	2.0 (0.4)	2.4 (0.9)
FOS-M6	363.6	2.7	0.9	1.4	N.D.	1.6
FOS-M7	467.2	N.D.	1.5	2.0	N.D.	2.3
FOS-M8	203.3	2.6	1.1	1.7	1.2	1.3
FOS-M9	275.8	3.8	N.D.	2.3	N.D.	1.4
Ave (SD)	327.5 (113.9)	3.0 (0.7)	1.2 (0.3)	1.9 (0.4)	1.2	1.7 (0.5)
GOS-M10	286.8	4.7	1.3	1.7	1.1	1.6
GOS-M11	358.7	3.1	1.3	1.1	0.6	1.8
GOS-M12	431.5	3.6	0.9	1.2	N.D.	1.0
GOS-M13	360.9	3.6	0.7	1.9	0.9	1.9
Ave (SD)	359.5 (59.1)	3.8 (0.7)	1.0 (0.3)	1.5 (0.4)	0.9 (0.3)	1.6 (0.4)
LS-M14	417.6	4.7	0.7	2.1	1.2	1.5
LS-M15	655.1	6.7	2.7	N.D.	N.D.	2.5
LS-M16	579.0	5.6	2.3	2.3	N.D.	2.5
LS-M17	425.8	4.2	1.9	2.4	N.D.	2.5
Ave (SD)	519.4 (117)	5.3 (1.1)	1.9 (0.9)	2.3 (0.1)	1.2	2.3 (0.5)

N.D. not detected.

## Data Availability

Not applicable.

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
