# Peer review of "Verification of the Impact of Blood Glucose Level on Liver Carcinogenesis and the Efficacy of a Dietary Intervention in a Spontaneous Metabolic Syndrome Model"

_ijms, 2021, doi:10.3390/ijms222312844_

Round 1

Reviewer 1 Report

All reviewer questions have been successfully answered.  

Author Response

To reviewer 1

Thank you very much for your comments.

Reviewer 2 Report

The article entitled "Verification of the impact of blood glucose level on liver carcinogenesis and the efficacy of a dietary intervention in a spontaneous metabolic syndrome model" has a big statistical problem. Again, it is not possible to use the % with a very low number of animals. It does not mean anything.

Table 2 is not correctly presented. The case "total" is a non-sense

Figure 4a: animals of 32 and 40 weeks of age cannot be summed and should be considered independantly

Table 5 is not understandable

It is possible that the hypothesis is not wrong but it would deserve more work with a greater number of animals.

Author Response

Dear Referee#2

Thank you very much for your comments. In particular, your good remarks on the statistical interpretation have helped us rethink the whole experimental design and reconfirm what is needed when translating the results of the mouse model to humans.

The deadline for this REVISION is one week, and we have revised the possible points within this period as follows. We would appreciate your consideration.

  1. The % presentation has been removed (Table 5).

(The % presentation of Experiment 1 is left as it is because of its large number )

  1. "Total" has been deleted (Table 2).

  1. The area of Langerhans islets has been showed independently at 32, 36 and 40 weeks of age (Figure 4A)

  1. The values in the table have been changed to the number of individuals, and the title was changed. A footnote was added (Table 5).

  1. We think that Experiment 1 was performed on a sufficient number of mice. However, as you pointed out, experiment 2 was performed on a small number of mice, and we did not find any apparent differences. Therefore, we understand that there is only a limited amount of evidence to discuss based on the results. We think it is essential to re-examine the results in a statistically meaningful number of mice to make the goal of translation to humans. On the other hand, we also found some disadvantages of the TSOD mice: they have a significant advantage as a spontaneous disease model, but they also have disadvantages due to their spontaneous nature (considerable individual variation and lack of severity of the disease). As it is not possible to represent the complex and diverse human disease landscape in a single animal model, we carefully considered the optimal design of additional experiments for the goal. We want to perform additional experiments using diet-induced mouse models that reproducibly express obesity, diabetes, and NASH with minimal individual variation. The present report is the first stage of verification based on the results of TSOD mice, and the content of discussion was limited to the proposal of a hypothesis based on the combination of the present results and previous reports.

Round 2

Reviewer 2 Report

the paper is better than the previous version. However, in several places (abstract and discussion sections) the text goes too far considering the lack of significant effects in the SCFA assays.

Author Response

Dear Reviewer #2

comments: the paper is better than the previous version. However, in several places (abstract and discussion sections) the text goes too far considering the lack of significant effects in the SCFA assays.

Thank you very much for your comments.

In the abstract, it was stated that milk-related oligosaccharides have cancer-preventive effects. In this revision, we tried to change the description after the result of the relationship between blood glucose level and carcinogenesis in experiment 1.

abstract L46-50

"Taken together, suppressing the increase in blood glucose level from a young age prevented susceptible individuals from diabetes and the onset of NAFLD/NASH, as well as carcinogenesis. Milk-derived oligosaccharides showed a lowering effect on blood glucose levels, which may be expected to prevent liver carcinogenesis."

In the Discussion, we explained too much about the involvement of SCFA from various angles, even though SCFA did not show significant results, which led to misunderstanding.
In the revision, we changed the description mainly in this part.

In Discussion, L 356-L383

"As for metabolic disturbance, improvements in fatty degeneration were observed in the oligosaccharide-treated groups. To unveil the underlying metabolic actions in the liver, we examined lipid-metabolism related genes such as Cpt-1a, Fasn, Fatp5, CD36, ChREBP, Mttp and glycogenesis-related genes including G6pase, Pepck1, Pepck2. However, all genes examined in the present study did not reveal the evidence of oligosaccharide-induced fatty liver suppression. One possibility for lacking significance between the treatment groups vs controls in this study was considered that the degree of hepatic fatty degeneration was too mild in the control group. As mentioned above, the advantage of TSOD mice is   their spontaneous development of various metabolic syndrome, however the disadvantage is the NASH-like pathogenic phenotypes less severe than other NASH models. Therefore, these metabolic genes in diet-induced NASH model will be further investigated.

Regarding the morphological changes in islets of Langerhans in the pancreas and their increased areas in hyperglycemic TSOD mice, it is likely that these changes reflect the compensatory swelling of beta-cells under insulin resistance; this is because the exocrine gland’s invasion into islets and beta-cell fragmentation clustering coincidently . Moreover, the degree of islet destruction was strongly correlated with the occurrence of liver tumors. Therefore, our results propose  a novel clinic perspective insofar as the risk of hepatic carcinogenesis could be predicted by the distortion degree of islets of Langerhans at a younger age. Based on the protective effects of milk-related oligosaccharides on beta-cells, we hypothesized that the potential underlying mode of action might be SCFAs produced from oligosaccharides by intestinal bacteria, but no significant changes in serum SCFA levels was observed. More sophisticate analysis on either the amount or organ-specific localization of SCFAs will be further investigated,  since beta-cells have SCFAs receptors (26,27), and several studies suggest the effect of propionic acid and butyric acid on improving insulin sensitivity and preventing inflammation. (28)(29) (30) (31)."